# Red Blood Cell Substitutes: Liposome Encapsulated Hemoglobin and Magnetite Nanoparticle Conjugates as Oxygen Carriers

**DOI:** 10.3390/ijms24021618

**Published:** 2023-01-13

**Authors:** Saleha Hafeez, Najam Us Sahar Sadaf Zaidi

**Affiliations:** Atta-ur-Rahman School of Applied Biosciences, National University of Sciences and Technology, H-12 Sector, Islamabad 44000, Pakistan

**Keywords:** RBC substitutes, molecular docking, magnetite nanoparticle, hemoglobin variants, protein corona, ion-exchange chromatography, liposome encapsulation

## Abstract

The established blood donation and transfusion system has contributed a lot to human health and welfare, but for this system to function properly, it requires a sufficient number of healthy donors, which is not always possible. Pakistan was a country hit hardest by COVID-19 which additionally reduced the blood donation rates. In order to address such challenges, the present study focused on the development of RBC substitutes that can be transfused to all blood types. This paper reports the development and characterization of RBC substitutes by combining the strategies of conjugated and encapsulated hemoglobin where magnetite nanoparticles would act as the carrier of hemoglobin, and liposomes would separate internal and external environments. The interactions of hemoglobin variants with bare magnetite nanoparticles were studied through molecular docking studies. Moreover, nanoparticles were synthesized, and hemoglobin was purified from blood. These components were then used to make conjugates, and it was observed that only the hemoglobin HbA1 variant was making protein corona. These conjugates were then encapsulated in liposomes to make negatively charged RBC substitutes with a size range of 1–2 μm. Results suggest that these RBC substitutes work potentially in a similar way as natural RBCs work and can be used in the time of emergency.

## 1. Introduction

Blood is an important life-sustaining fluid, a special type of connective tissue that circulates throughout the body to achieve three main functions: transportation, protection, and regulation. These functions are accomplished by two of its main components: plasma and formed elements. Plasma is a protein-rich fluid in which formed elements such as red blood cells, platelets, and white blood cells are suspended [1,2,3]. In recent years, blood transfusion is mainly performed by the transfusion of whole blood (limited to massive resuscitation or trauma situations) or any other component of the blood [4]. Packed red blood cells from the blood are the most commonly used component to save patients’ lives via hemoglobin, an oxygen-carrying metalloprotein [5].

Every year about 118.5 million blood donations are globally collected. With a properly established hemovigilance system, blood donation and transfusion are now considered safe and reliable in high-income developed countries, which account for only 16% of the world’s population. However, in low-income developing countries, there is always a shortage of blood supply due to a lack of healthy volunteers [6]. According to Regional Status Report on Blood Safety and Availability 2016 for the Eastern Mediterranean published in 2017 [7], Pakistan is among those low-income countries where only 10% of the required blood donations are from volunteers. The rest 90% of the collection comes from families’ replacement donations which are often used to treat pregnancy-related complications and severe anemia in children. In addition to having a fragmented hemovigilance system of blood banks, ongoing pandemic lockdowns have added more fuel to the fire, and now the whole country is struggling to maintain a sufficient blood supply as the blood donation rate has been reduced and most affected are the thalassemic patients. Pakistan has over 100,000 thalassemia patients, each requiring 3 to 4 pints of fresh blood per month in the form of packed RBCs [8]. A study conducted on the wastage of blood products in Pakistan by Mujtaba et al. (2020) observed that all packed RBCs were fully utilized, and there was still a shortage of supply for transfusion [9].

Due to the increased demand for blood for transfusion and concerns related to blood incompatibility, lack of voluntary donors, bloodborne pathogens, and short shelf life, several companies are working on the production of safe and effective hemoglobin-based oxygen carriers (HBOCs). Although many steps have been taken to make ideal RBCs substitutes, no hemoglobin-based oxygen carrier as a substitute has been approved by the FDA due to various reasons [5]. First and second generations of hemoglobin-based substitutes were focused on the transfusion of cell-free hemoglobin that would carry gases the same way as hemoglobin works but in a cell-free environment. These substitutes were cross-linked (linked recombinant hemoglobin), polymerized (linked with glutaraldehyde), and conjugated (PEG) to overcome the problems associated with free hemoglobin dissociation. Due to unexpected toxicity and high mortality rates, these HBOCs were discontinued. The concept of third-generation HBOCs was to make an artificial cell encapsulating proteins the same way as RBC encapsulates hemoglobin and enzymes. These RBCs substitutes were aimed at mimicking some of the essential features of RBCs [10,11].

In the present study, we aimed to combine the strategies of conjugated hemoglobin and encapsulated hemoglobin in order to develop RBC substitutes that would carry hemoglobin as an oxygen carrier, magnetite nanoparticles conjugated with hemoglobin (as a carrier of hemoglobin and to prevent the rapid release of hemoglobin into the bloodstream) and liposomes to shield the internal environment from external.

## 2. Results and Discussion

### 2.1. Docking of Hemoglobin with Bare Magnetite Nanoparticles

Molecular docking studies were carried out to predict the interactions of different hemoglobin variants with bare magnetite nanoparticles. In this study, instead of using a complete spherical nanoparticle, we used a slab to represent a portion of the surface of the whole nanoparticle. Blind docking was performed to allow the slab to interact anywhere with the hemoglobin molecule. Based on the results, only hemoglobin variants HbA1 and HbF were found interacting with the slab, and binding energies were −20.4 kcal/mol and −11.6 kcal/mol, respectively. These binding energies were the same in all generated poses. It was observed that in all generated poses, both chains of hemoglobin variants interacted with the slab. In the case of beta and gamma chains, only the beta chain showed more interactions than the gamma chain (Figure 1 and Appendix A
Figure A2). Results of molecular docking of all three hemoglobin variants are summarized in Table 1, and 2D diagrams of the first binding poses showing the interactions of HbA1 and HbF are shown in Figure 2 and Appendix A
Figure A3, respectively. The overall result of docking indicates that hemoglobin molecules have the ability to interact with bare magnetite nanoparticles and form a protein corona around them. Similar molecular docking studies involving the interaction of a nanoparticle and protein have been reported by various research groups [12,13,14].

### 2.2. Production of Bare Magnetite Nanoparticles

In the present study, we introduced a cheaper method of synthesizing bare magnetite nanoparticles by using lab-produced carbon dioxide gas to evacuate oxygen, as mentioned in Appendix A. Usually, nitrogen and argon gases are used to evacuate oxygen from the environment due to their inert properties [15,16], but a continuous supply of gas was required throughout the experiment which was not feasible using the available protocols.

The obtained bare magnetite nanoparticles produced by the chemical co-precipitation synthesis method were carefully characterized through various techniques (result images shown in Appendix A
Figure A4). Scanning Electron Microscope (SEM) technique was used to determine the morphology and size of crystallite. SEM micrographs showed irregularly shaped and aggregated nanoparticles. A heterogeneous population of nanoparticles ranging from 19 nm to 45 nm was observed. Energy Dispersive X-Ray Analysis (EDX) of synthesized MNPs showed more than 98% purity of Fe and O elements which were slightly higher than that of other reported methods [15,17]. Less than 2% was contributed by contaminants such as sodium (Na) and chloride (Cl). X-Ray diffraction technique was further used to study the crystalline structure and to determine the phases of nanoparticles. XRD pattern of bare magnetite nanoparticles showed crystalline nature with six strong and sharp peaks at 2θ: 30.2, 35.5, 43.2, 53.5, 57.1, and 62.9 corresponding to crystalline planes: (220), (311), (400), (422), (511) and (440) of magnetite phases, respectively (magnetite JCPD card no: 19-629). UV-VIS spectroscopy was used to study the optical properties of nanoparticles. UV-VIS spectrum of bare magnetite nanoparticles revealed a characteristic absorption peak of magnetite nanoparticles at a wavelength of 375 nm, which is within the characteristic wavelength range of magnetite nanoparticles (330–450 nm) observed by another research group [18]. The Zeta potential of bare magnetite nanoparticles at pH 7.4 in distilled water was revealed to be −14.1 mV. The FTIR spectrum obtained from as-synthesized bare magnetite nanoparticles exhibited a vibrational band at 580 cm^−1,^ characteristic of magnetite nanoparticles (Figure 5). This band corresponds to Fe-O bonds in the crystallite lattice of Fe_3_O_4_, as reported by Nalbandian et al. (2015) [19].

### 2.3. Preparation of Deoxyhemoglobin Stock Solution

Ultra-purification of hemoglobin has been performed by several groups [20,21,22,23]. Hemoglobin was isolated from the blood through hemolysis of RBCs, a method reported by Russo and Sorstokke (1973) [24]. Figure 3 shows the SDS PAGE of unpurified and purified hemoglobin (an image of the chromatogram is provided in Appendix A
Figure A5).

Different hemoglobin variants were detected through cation exchange chromatography in impure hemoglobin solution, as shown in Table 2. The retention times of all variants were similar to the retention times mentioned by Szuberski, Oliveira, and Hoyer (2012) [25]. However, after purification, only HbA1 was observed. The possible explanation for the appearance of only HbA1 in the purified sample is that HbA1 is present in high concentrations (95%) in normal blood, whereas the concentrations of other normal hemoglobin variants HbF and HbA2 are very low (<1% and 2.2–3.5% respectively) [26], hence HbF and HbA2 must have eluted along with other contaminants and did not appear in the purified sample or the concentrations in the purified sample were too low to appear in the chromatogram (image of the chromatogram in Appendix A
Figure A7 and Figure A8). FTIR spectrum, as shown in Figure 5 of a purified aqueous solution of hemoglobin, exhibited vibrational bands at 1652 cm^−1^ (amide I) and 1546 cm^−1^ (amide II). Other bands 3450 cm^−1^, 1395 cm^−1^, 1164 cm^−1^, 1131 cm^−1^ and 667 cm^−1^ corresponds to the O-H, C = O, N-H, C-H, C = O and C = O functional groups respectively. Similar results have been reported by another research group [27].

### 2.4. Preparation of Deoxyhemoglobin-Magnetite Conjugates

Cell-free hemoglobin as substitutes were unsuccessful because of renal toxicity caused by free hemoglobin [10,11]. Free hemoglobin dissociates into subunits, and one of the factors which enhance the dissociation is low hemoglobin concentration [28]. The main advantage of using magnetite nanoparticle is that it acts as a carrier where hemoglobin is concentrated on the surface in the form of protein corona; hence it will prevent dissociation. The additional advantage is that magnetite nanoparticles are made up of iron which will ultimately become part of iron pool of the body.

In order to identify the type of hemoglobin variants making protein corona, unbound hemoglobin and hemoglobin corona were removed and confirmed by SDS PAGE. SDS PAGE showing removed unbound hemoglobin is shown in Appendix A
Figure A9. Removal of both soft and hard hemoglobin corona is shown in Figure 4.

In the FTIR spectrum (Figure 5) of conjugates, the interaction of hemoglobin with magnetite nanoparticles was confirmed by bands appearing at 3443 cm^−1^, and 1631 cm^−1^, which were from hemoglobin and bands appearing at 2918 cm^−1^, 2850 cm^−1^, and 582 cm^−1^ were from magnetite nanoparticles. The appearance of the vibrational band at 582 cm^−1^ confirmed the presence of a magnetite core, and bands appearing at 3443 cm^−1^, and 1631 cm^−1^ were amide A and amide I, confirming the presence of hemoglobin corona. The effective charge on the surface of conjugates at pH 7.4 in blood plasma was revealed to be −17.8 mV.

Since only HbA1 was present in the purified deoxyhemoglobin solution, we had to make conjugates using impure hemoglobin solution to check which hemoglobin variant would prefer to interact with bare magnetite nanoparticles. Based on the chromatogram obtained through cation exchange chromatography (Appendix A
Figure A8), only HbA1 was detected in the conjugates making hemoglobin corona (Table 2). However, the results of molecular docking indicated the interaction of hemoglobin variants HbA1 and HbF with bare magnetite nanoparticles. The explanation for the appearance of only HbA1 lies in the concentration of hemoglobin variants. The isoelectric point of HbA1, HbF, and HbA2 are 6.9, 7.0, and 7.2, respectively [29], which means that at physiological pH 7.4 all variants of HbA1, HbF, and HbA2 would have a negative charge, but in our case, the charge had little or no influence on corona formation because of very high concentrations of HbA1 in the sample, which was able to make corona around nanoparticles. Even if other variants were interacting with nanoparticles, these were not detected in hemoglobin corona due to very low concentrations.

### 2.5. Analysis of the Presence of Hemoglobin in Blood Plasma from Conjugates and Presence of Blood Plasma Proteins on Conjugates

The first environment that nanoparticles meet when injected intravenously into the body is the environment of the blood, which is composed of different types of components, including cells, proteins, lipids, and various different molecules. These components have the ability to interact with nanoparticles. Proteins that have an affinity for nanoparticles form protein corona [30]. According to Vroman’s effect, the highest mobility proteins with low affinity are slowly replaced by high-affinity proteins present in low concentrations [31]. To study the fate of conjugates in blood, we had to check whether plasma proteins have the ability to replace adsorbed hemoglobin and also check whether conjugates were releasing hemoglobin into the blood. As shown in Figure 6, hemoglobin started detaching (lanes 2–7) soon after conjugates faced the blood plasma environment, but after several hours of constant washing and shaking, no hemoglobin was detected in the plasma (lanes 8 and 9). Here we assumed that all the loosely attached hemoglobin, i.e., soft hemoglobin corona, was removed and replaced by newly formed protein corona consisting of blood plasma proteins. However, when the composition of protein corona was checked after incubation of conjugates in blood plasma, only hemoglobin was found attached to nanoparticles, as shown in Figure 7. The simplest explanation is that since hemoglobin was actually covering the surface of magnetite nanoparticles, blood plasma proteins did not find space to interact directly with nanoparticles, and hence the only possible interaction was to form soft protein corona through protein-protein interactions, but in this case, we also did not observe soft protein corona made of blood plasma proteins. This could be due to the repulsion between the negatively charged conjugates (−17.8 mV) and negatively charged blood plasma proteins such as immunoglobulins [32] and albumin [33], and hence no interactions were observed. Adsorption of plasma proteins mainly depends on the surface charge and hydrophobicity of nanoparticles [34,35]. Various research groups have studied the interactions of plasma proteins and nanoparticles [34,36].

### 2.6. Development of RBC Substitutes

In order for RBC substitutes to work in a similar way as natural RBCs work in blood, substitutes must mimic some of the features of natural RBCs, i.e., substitutes must have a: 1. high hemoglobin content, 2. large in size and 3. strong negative charge.

The composition of liposomes directly affects the stability and encapsulation efficiency. The higher the cholesterol content is, the more stable the liposomes are, and the lesser the encapsulation efficiency these have [37]. In the present study, the encapsulation efficiencies were 40%, 45%, 60%, and 65% for 1:4, 1:6, 1:8, and 1:10 ratios, respectively. These results are in agreement with the results published by Colletier et al. (2002) [38], who reported that the encapsulation efficiency of a protein increases with the increase in the lipid content of liposomes, and encapsulation depends on the interaction between protein and the lipid.

Unlike natural RBCs, which have the ability to deform due to their biconcave shape^2^, liposomes are spherical in shape [39]. In order to prevent substitutes from crossing blood-brain barrier [40], these substitutes must have a large size (preferably in μm range) but should be smaller than 5μm to prevent them from clogging the microcapillaries which would compromise the blood flow [41]. In this study, we managed to produce RBC substitutes within the range of 1–2μm, which was confirmed through transmission electron microscopy, as shown in Figure 8.

In order to prevent RBC substitutes from interacting with each other and with blood components, substitutes must possess a strong negative charge because all biological membranes and blood plasma proteins bear a net negative charge, including RBC’s membrane [42]. RBCs have sialylated glycoproteins on their membrane, which are responsible for the negative surface charge. This negative charge creates repulsive electric zeta potential between cells [43]. In the present research, negative charges on substitutes in distilled water and plasma were revealed to be −43.32 mV and −31.41 mV, respectively. Several studies have been conducted to determine how liposomes interact with plasma proteins and lipoproteins [42,44,45,46].

## 3. Materials and Methods

### 3.1. Materials, Software, and Apparatus

Software used for molecular docking was Avogadro version 1.2, ArgusLab version 4.0.1, Biovia Discovery Studio Visualizer 2021, and PyRx version 0.8. All the chemicals and reagents used were of analytical grade and used without further purification. Fresh screened blood was provided by NUST ASAB Diagnostics Lab. Purification of hemoglobin was performed by Next Generation Chromatography NGC Quest 100 plus chromatography system by Bio-Rad. SDS PAGE was performed on protein gel electrophoresis equipment by Bio-Rad. Nanoparticles were dispersed by ultrasonicator UP400S (Hielscher Ultrasound Technology). Scanning electron micrography was performed by VEGA3 TESCAN, and transmission electron microscopy was performed by JEOL JEM 1010. UV-VIS spectroscopy was performed by UV-VIS spectrophotometer (Agilent Technology). X-ray diffraction patterns were obtained by an X-ray diffraction analyzer (Bruker). Zeta potential was recorded by a zetasizer equipped with a HeNe laser (Malvern Zetasizer Nano ZS). Fourier-transform infrared spectroscopy spectra were obtained from an FTIR spectrometer (PerkinElmer—Spectrum 100 FTIR Spectrometer). Different types of hemoglobin variants were identified by Bio-Rad variant 2. Liposomes were prepared by rotary evaporator (InLabo, Berlin, Germany).

### 3.2. Docking of Hemoglobin with Bare Magnetite Nanoparticles

Molecular docking was performed to study the interaction of different hemoglobin variants with bare magnetite nanoparticles. A slab of Fe_3_O_4_ (10 nm × 10 nm × 10 nm) was made using Avogadro software [47]. Crystal structures of normal hemoglobin variants (HbA1 PDB ID: 1KD2, HbA2 PDB ID: 1SI4, HbF PDB ID: 1FDH) were obtained from Protein Data Bank (PDB). Universal Force Field (UFF) was used for both geometry and energy optimization by combined use of ArgusLab [48] and Avogadro software. Molecular docking was performed on PyRx [49] with grid parameters set to maximum. Results obtained from PyRx were analyzed on Biovia Discovery Studio Visualizer 2021 [50].

### 3.3. Production of Bare Magnetite Nanoparticles

Bare magnetite nanoparticles were synthesized by the traditional chemical co-precipitation method, as mentioned by Yazdani and Seddigh (2016) [15], with slight modifications. Analytical grade ferric and ferrous sources were dissolved in deionized water (150 mL) in a ratio of 2:1 (1 M Ferric Chloride Hexahydrate FeCl_3_.6H_2_O and 0.5 M Ferrous Sulfate Heptahydrate FeSO_4_.7H_2_O) respectively at room temperature. The solution was stirred at a constant speed of 500 rpm and kept under a carbon dioxide atmosphere to prevent oxidation. The production of carbon dioxide gas is explained in Appendix A (Figure A1). After a few minutes of constant stirring, 30 mL of alkaline agent (1 M NaOH) was added dropwise to precipitate the nanoparticles. Nanoparticles were magnetically separated and washed 3 to 4 times with absolute ethanol to remove any unreacted precursors. After multiple washings, the nanoparticles were dried in a vacuum oven at 80 °C for 3 h.

### 3.4. Preparation of Deoxyhemoglobin Stock Solution

A stock solution of deoxyhemoglobin was prepared from fresh screened human blood. RBCs were separated and washed with 0.9% normal saline solution several times. Hemolysis of the cells was performed by adding 1.2 volumes of water, and lipids were extracted in 0.4 volumes of toluene. The mixture was shaken vigorously and incubated at room temperature for an hour. After incubation, the toluene layer was removed, and the mixture was centrifuged at the highest speed. Cell debris was removed, and the resulting aqueous solution was the stock solution of oxyhemoglobin [24]. Hemoglobin was purified by anion exchange chromatography. Protein contaminants such as enzymes, cell membrane proteins, surface antigens, and any other contaminants were removed during the purification step. After purification, types of normal hemoglobin variants (adult hemoglobin variants HbA1, HbA2, and fetal hemoglobin HbF) in purified and unpurified hemoglobin solution were identified by cation exchange chromatography. The hemoglobin solution was concentrated to 3 g/mL (quantified by Bradford Assay) by using a centrifuge concentrator plus, and finally, it was converted to deoxyhemoglobin by the drop-wise addition of Stokes reagent according to a method mentioned by Russo and Sorstokke (1973) [24].

### 3.5. Preparation of Deoxyhemoglobin-Magnetite Conjugates

Deoxyhemoglobin and magnetite nanoparticle conjugates were prepared by dispersing 2 mg nanoparticles in a 5 mL stock solution of purified deoxyhemoglobin (3 g/mL) and in an impure solution of deoxyhemoglobin by using an ultrasonicator at 60% amplitude and 0.5 cycles for 45 min in the ice box. After sonication, the conjugates were incubated at 4 °C for 30 min to 1 h. After incubation, unbound hemoglobin was removed by washing several times with distilled water which was confirmed by SDS PAGE [51]. Final conjugates were characterized through various techniques and were divided equally for two different experiments: 1. types of hemoglobin variants making protein corona, 2. development of RBC substitutes.

#### 3.5.1. Fourier-Transform Infrared Spectroscopy (FTIR)

In order to confirm the presence of hemoglobin corona around bare magnetite nanoparticles, an infrared spectrum of absorption of bare nanoparticles (solid), purified hemoglobin (liquid), and nanoparticle-hemoglobin conjugates (solid) was obtained using an FTIR spectrometer. Powdered nanoparticles and conjugates were compressed separately with KBr to form pellets, and a drop of hemoglobin was dropped on the compressed KBr pellet. The FTIR spectra were recorded in transmittance mode within the wavenumber range of 500–4000 cm^−1^.

#### 3.5.2. Zeta Potential Analysis

Charge on bare magnetite nanoparticles and conjugates of nanoparticle-hemoglobin was determined by zetasizer. Bare nanoparticles and conjugates were dispersed (1 mg/mL) in distilled water maintained at pH 7.4 and blood plasma, respectively. Samples were diluted till the final concentration of nanoparticles, and conjugates were 10 ug/mL. For analysis, samples were run in an automatic mode (temperature set at 37 °C) using zeta potential dip cell ZEN1002.

Types of hemoglobin variants interacting with bare magnetite nanoparticles were identified by cation exchange chromatography. Bound hemoglobin corona was removed by washing with increasing concentrations of NaCl solution (0.5 M–2 M), and removal was confirmed by SDS PAGE. Dialysis was performed to remove NaCl by incubating a dialysis tube (14 kDa MWCO) containing removed hemoglobin corona placed in continuously stirring distilled water. Hemoglobin variants were identified by cation exchange chromatography and compared with purified and unpurified hemoglobin solutions.

### 3.6. Analysis of the Presence of Hemoglobin in Blood Plasma from Conjugates and Presence of Blood Plasma Proteins on Conjugates

In this part, we studied the release of hemoglobin from conjugates and the interaction of conjugates with blood plasma proteins. To check the presence of hemoglobin in plasma, we washed the conjugates with blood plasma for several hours (15 min to 6 h) under constant shaking at 37 °C. After each washing, the plasma was replaced with fresh plasma. To check whether plasma proteins were interacting with nanoparticles, we washed the conjugates with distilled water after 6 h of constant shaking. The presence of hemoglobin in plasma and the presence of plasma proteins on nanoparticles were confirmed by SDS PAGE.

### 3.7. Development of RBC Substitutes

RBC substitutes were prepared by the conventional thin film hydration method [52,53,54] in ratios of 1:4, 1:6, 1:8, and 1:10 of cholesterol and lecithin by taking a quantity of lecithin (50 mg) constant and varying the concentrations of cholesterol (12.5 mg, 8.3 mg, 6.25 mg, and 5 mg respectively). All of these ratios were prepared by dissolving lecithin and cholesterol in 10 mL chloroform, followed by vacuum evaporation at 60 °C for 45 min in a rotary evaporator. Hydration was performed by the solution of hemoglobin and nanoparticle conjugates under constant shaking at 120 rpm in a shaking incubator. The mixture was incubated for 15 min at room temperature. Encapsulation efficiencies of different ratios were compared, and the one with the highest encapsulation efficiency was selected. RBC substitutes were extruded multiple times through filter paper pore size 3 μ. Most of the unencapsulated conjugates were removed during this step, and the rest were removed by centrifugation. During each round of centrifugation, the supernatant was removed and replaced with fresh blood plasma. A red pellet was obtained (Figure 8), which was stored at 4 °C. This pellet was then characterized through transmission electron microscopy and zeta potential analysis.

#### 3.7.1. Transmission Electron Microscopy (TEM)

Transmission electron microscopy was performed to determine the size of RBC substitutes and to confirm the encapsulation of conjugates. The substitutes were first washed several times with phosphate buffer and centrifuged at high speed of 5000 rpm for 10 min to obtain a red pellet. After washing, substitutes were first fixed for 2 h in primary fixatives (2.5% Glutaraldehyde and 4% p-Formaldehyde in 0.2 M Phosphate buffer). Then, the substitutes were washed several times with the same buffer for 10 min and fixed for 30 min in a secondary fixative (4% Osmium Tetroxide OsO_4_). After secondary fixation, the substitutes were washed several times, first with phosphate buffer and second with distilled water for 10 min. A drop of substitutes was placed on the TEM grid (copper grid) and allowed to adsorb. Excessive water was removed with filter paper. A drop of 5% uranyl acetate was left in contact with the substitutes for 5 min. Excessive stains and water were removed with filter paper. The substitutes were allowed to dry at room conditions before they were sent to the electron microscope facility.

#### 3.7.2. Zeta Potential Analysis

The effective charge on the surface of RBC substitutes was determined by zetasizer. Samples were prepared by incubating 10 uL of substitutes in 1 ml distilled water set at pH 7.4 and 1 mL blood plasma at 37 °C for 1 h under constant shaking, for analysis samples, were run in an automatic mode (temperature set at 37 °C) using zeta potential dip cell ZEN1002.

## 4. Conclusions

The aim of this study was to develop RBC substitutes that will mimic red blood cells structurally and functionally by encapsulating conjugates of MNPs-hemoglobin in liposomes. These substitutes have the potential to work in a similar way as RBCs work and can be transfused to any blood type in the form of packed RBC substitutes, such as in the case of thalassemic patients and severely anemic patients. However, these substitutes are not suitable in situations where massive blood transfusions are required. The current study focuses only on the development. More research is required to (i) study the interaction of whole RBC substitute with blood proteins, (ii) further stabilize the conjugates, (iii) functionally analyze the substitutes, and (iv) improve circulation in the circulatory system.

## Figures and Tables

**Figure 1 ijms-24-01618-f001:**
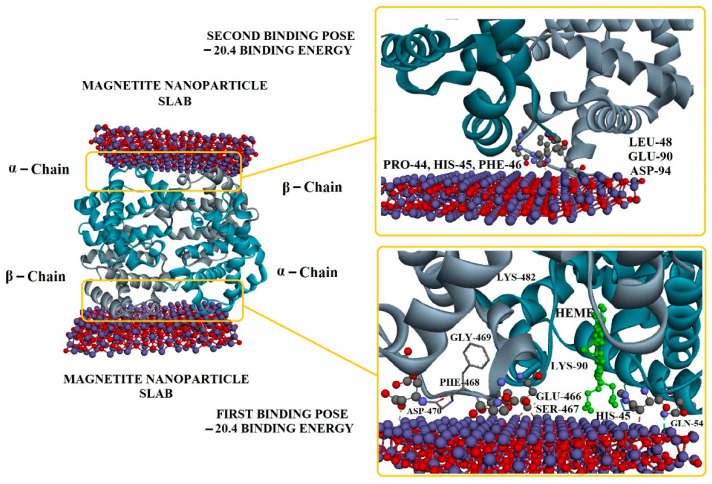
Interaction of hemoglobin HbA1 with bare magnetite nanoparticle slab.

**Figure 2 ijms-24-01618-f002:**
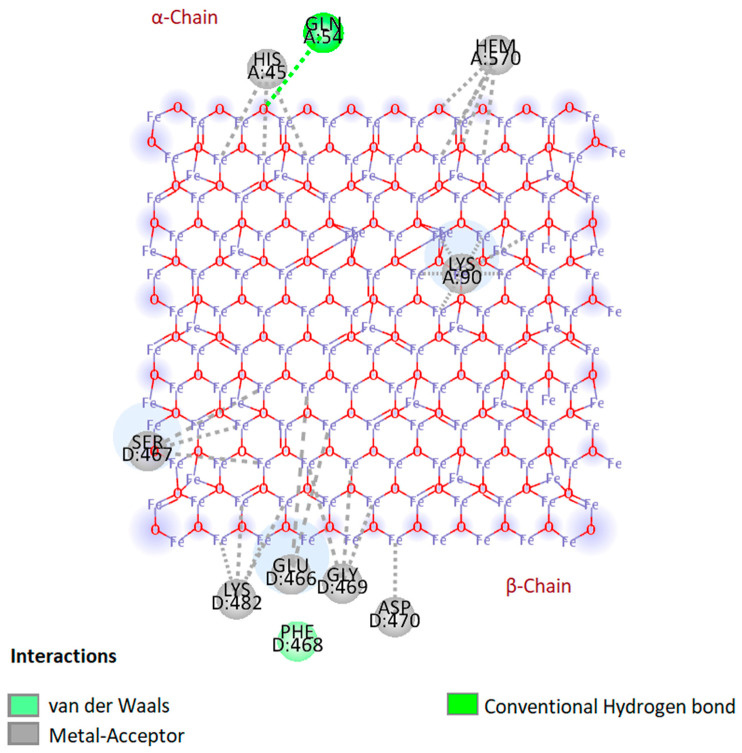
Two-dimensional (2D) diagram showing interactions of hemoglobin HbA1 with bare magnetite nanoparticle slab in the first binding pose.

**Figure 3 ijms-24-01618-f003:**
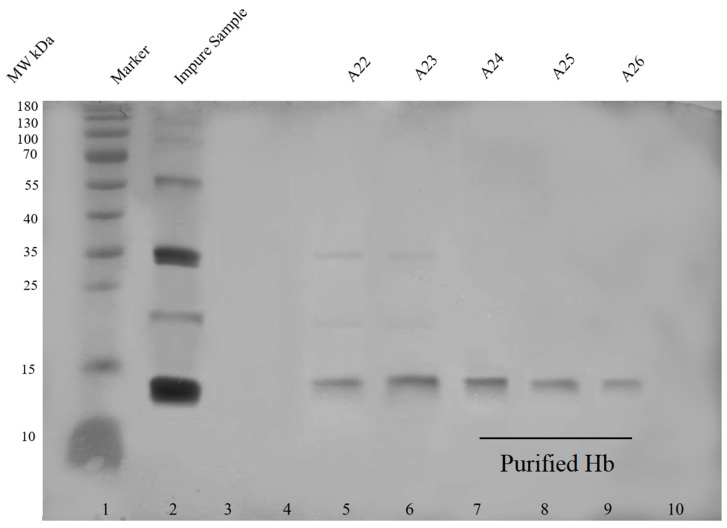
Purification of hemoglobin through anion exchange chromatography. Silver stained SDS PAGE gel (12%) (uncropped single gel) showing hemoglobin before and after purification. The impure sample showed multiple bands of impurities. Impurities were also observed in fractions A22 and A23. Hemoglobin was completely purified in fractions A24–A26. The color image of the complete gel is shown in Appendix A
Figure A6.

**Figure 4 ijms-24-01618-f004:**
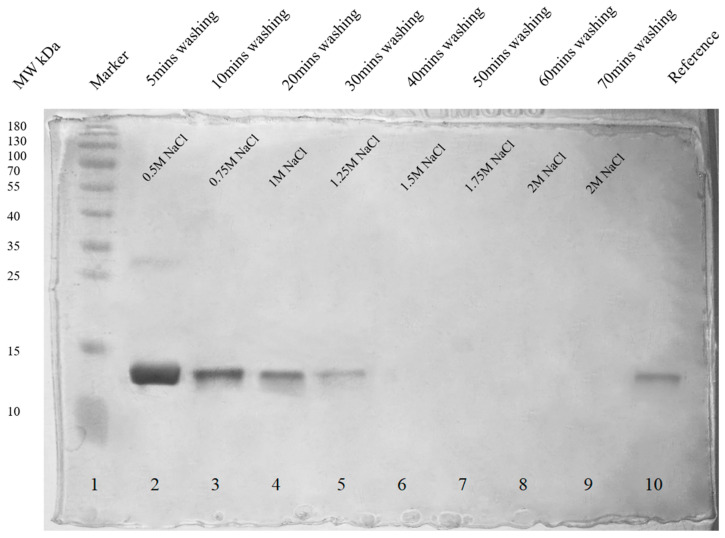
SDS PAGE (12% silver-stained uncropped gel) analysis of removed hemoglobin corona. Lane 2 shows the maximum removal of hemoglobin corona after washing with 0.5 M NaCl for 5 min. Lane 3 to 9 shows the removal of hemoglobin after washing with gradually increasing concentrations of NaCl (0.75 M, 1 M, 1.25 M, 1.5 M, 1.75 M, and 2 M), and no protein was detected after 30 min of washing as shown in lane 6. Lane 10 is the reference hemoglobin protein. The color image of uncropped gel is shown in Appendix A
Figure A10.

**Figure 5 ijms-24-01618-f005:**
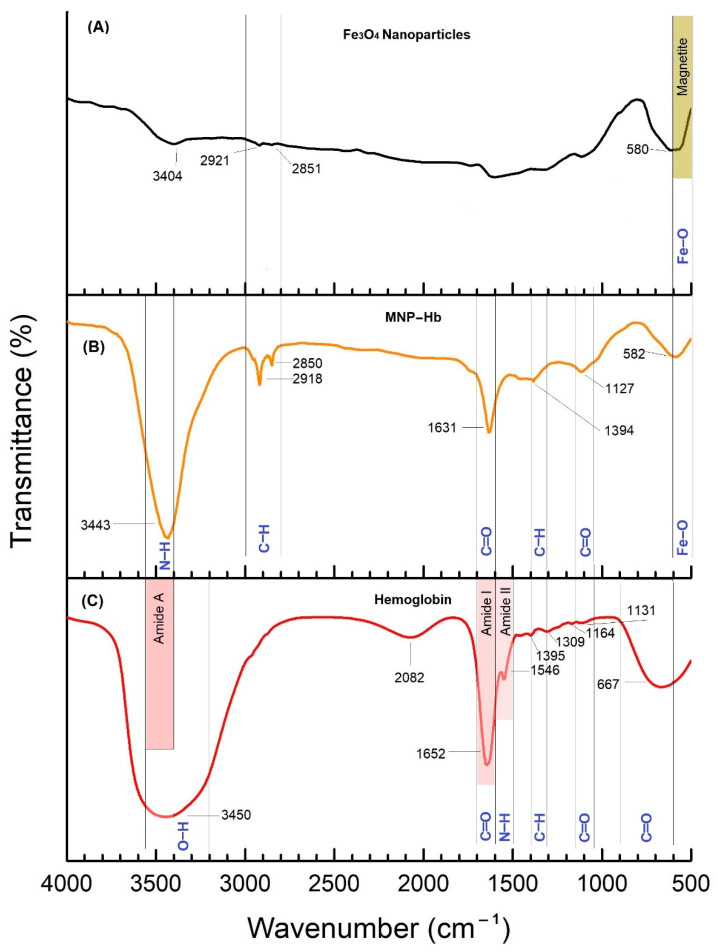
FTIR spectra of (**a**) bare magnetite nanoparticles, (**b**) hemoglobin and magnetite conjugates, (**c**) Hemoglobin. Note: MNP, MNP-Hb, and Hb represent magnetite nanoparticles, magnetite nanoparticles-hemoglobin conjugates, and hemoglobin, respectively.

**Figure 6 ijms-24-01618-f006:**
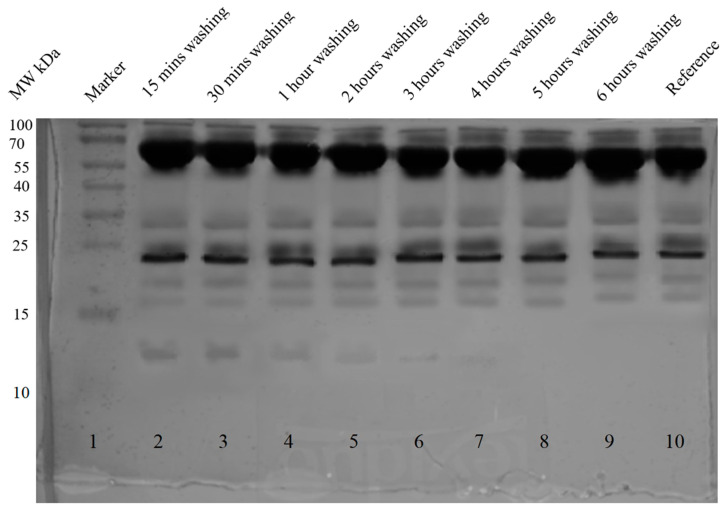
SDS PAGE (15% silver-stained uncropped gel) analysis of the presence of hemoglobin in blood plasma. Lanes 2 to 7 show the presence of hemoglobin in plasma from conjugates. Lanes 8 and 9 show no presence of hemoglobin in plasma from conjugates. Lane 10 is reference blood plasma.

**Figure 7 ijms-24-01618-f007:**
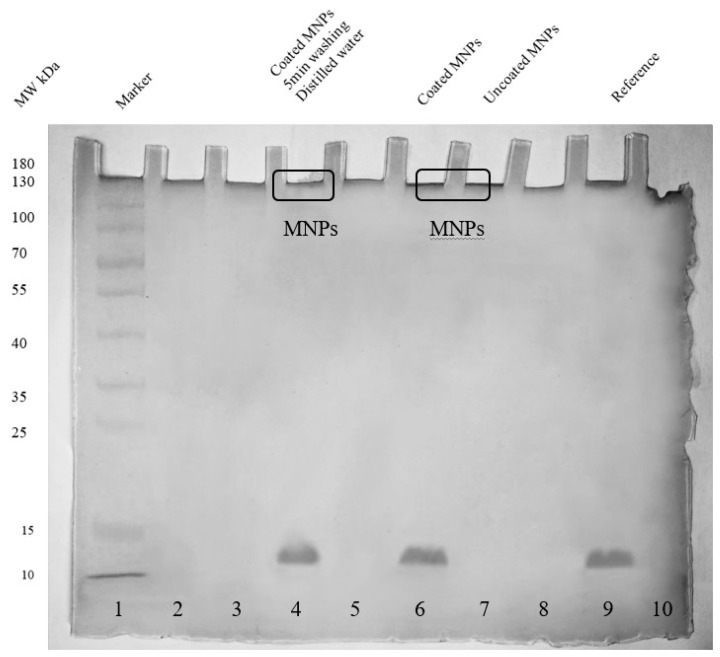
SDS PAGE (15% silver-stained uncropped gel) analysis of the presence of blood plasma proteins on conjugates. Lane 4 shows no presence of blood plasma proteins on conjugates. Conjugates of magnetite nanoparticles-hemoglobin and uncoated magnetite nanoparticles were taken as control, as shown in lanes 6 and 7, respectively. Lane 9 shows reference hemoglobin molecules. The color image of uncropped gel is shown in Appendix A
Figure A11.

**Figure 8 ijms-24-01618-f008:**
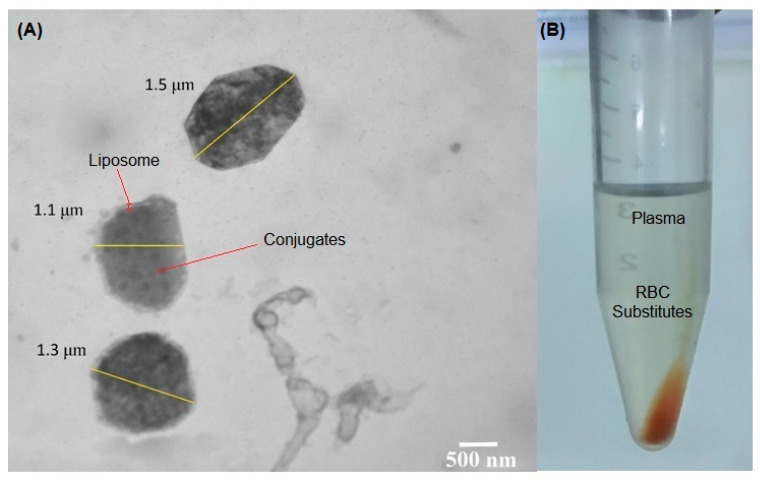
(**A**) TEM image of RBC substitutes. The coat represents liposome, the core represents the conjugates of magnetite and hemoglobin, (**B**) Pellet of RBC substitutes in blood plasma.

**Table 1 ijms-24-01618-t001:** Molecular docking results of different hemoglobin variants.

Hemoglobin Variants	Binding Energy kcal/mol	Interacting Residues	Interaction Types
**HbA1**	−20.4 (1st Binding Pose)	**Chain Alpha:**	
		GLN-54	Conventional Hydrogen Bond
		HIS-45	Metal-Acceptor
		LYS-90	Metal-Acceptor
		HEME	Metal-Acceptor
		**Chain Beta:**	
		GLU-466	Metal-Acceptor
		SER-467	Metal-Acceptor
		PHE-468	van der Waals
		GLY-469	Metal-Acceptor
		ASP-470	Metal-Acceptor
		LYS-482	Metal-Acceptor
	−20.4 (2nd Binding Pose)	**Chain Alpha:**	
		PRO-44	Metal-Acceptor
		HIS-45	Metal-Acceptor
		PHE-46	Metal-Acceptor
		**Chain Beta:**	
		LEU-48	Carbon-Hydrogen Bond
		GLU-90	Metal-Acceptor
		ASP-94	Metal-Acceptor
**HbA2**	0.0		
**HbF**	−11.6 (1st Binding Pose)	**Chain Alpha:**	
		HIS-327	Metal-Acceptor
		ASP-329	Metal-Acceptor
		GLN-336	Metal-Acceptor
		LYS-372	Metal-Acceptor
		**Chain Gamma:**	
		ASP-184	Metal-Acceptor
		SER-185	Metal-Acceptor
		LYS-236	Conventional Hydrogen Bond
	−11.6 (2nd Binding Pose)	**Chain Alpha:**	
		HIS-45	Metal-Acceptor
		ASP-47	Metal-Acceptor
		GLN-54	Metal-Acceptor
		LYS-90	Metal-Acceptor
		**Chain Gamma:**	
		ASP-467	Metal-Acceptor

**Table 2 ijms-24-01618-t002:** Identification of hemoglobin variants through cation exchange chromatography.

Sample	Variants	Retention Time (min)
**Impure Hemoglobin**	HbF, HbA1c, HbA1, HbA2	1.13, 1.54, 2.46 and 3.60
**Purified Hemoglobin**	HbA1	2.44
**Conjugates of Impure Hb and MNPs**	HbA1	2.44
**Conjugates of Purified Hb and MNPs**	HbA1	2.43

## Data Availability

Not applicable.

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
