# Peer review of "Red Blood Cell Substitutes: Liposome Encapsulated Hemoglobin and Magnetite Nanoparticle Conjugates as Oxygen Carriers"

_ijms, 2023, doi:10.3390/ijms24021618_

Round 1
Reviewer 1 Report
In their manuscript "Red Blood Cell Substitutes: Liposome Encapsulated Hemoglobin and Magnetite Nanoparticle Conjugates as Oxygen Carriers" the authors provide interesting information regarding the production of red blood cell substitutes for transfusion purposes. I really enjoyed the workflow and the study is well designed. I only have some minor comments:
1) In the Introduction section the authors mention "blood incompatibility, lack of voluntary donors, bloodborne pathogens and short shelf life". I believe it would be important to also mention storage lesion and donor variation effect upon transfusion, since red blood cell substitutes can help address those challenges too.
2) I believe that the manuscript needs a conclusion section to collect in brief the idea and the results presented, as well as to highlight the impact of this innovation upon transfusion therapy and its future perspectives.
Author Response
1) In the Introduction section the authors mention "blood incompatibility, lack of voluntary donors, bloodborne pathogens and short shelf life". I believe it would be important to also mention storage lesion and donor variation effect upon transfusion, since red blood cell substitutes can help address those challenges too.
Yes, we agree that it would be important to mention storage lesion and donor variation effect upon transfusion, since red blood cell substitutes can help address those challenges too. In Pakistan blood banks are mostly run by various communities and non-governmental organizations (NGOs) and face acute shortage of supply due to lack of voluntary donations. Due to this, products of blood such as packed RBCs are fully consumed on the same day that they are received. Thus, the issue of shortage persists. Currently storage lesion and related donor variation effects upon transfusion is not an issue in Pakistan (unfortunately).
2) I believe that the manuscript needs a conclusion section to collect in brief the idea and the results presented, as well as to highlight the impact of this innovation upon transfusion therapy and its future perspectives.
Worthy reviewer, we have added a conclusion section. The main aim was to produce RBC substitutes using less and available resources. We have used the cheapest possible ways to produce RBC substitutes which may be the best option in countries where blood related emergencies are high due to low donation rates, poor infrastructural management, and low resources.
Reviewer 2 Report
The authors presented an interesting manuscript on a topical issue. However, the form of presentation of the material makes it difficult for the reader to perceive it. For this reason, several additions and changes must be made to the manuscript.
1. In the introduction, the current state is practically not discussed, but an analysis of publications devoted to the design of HBO is necessary.
2. The authors need to substantiate the choice of the way to solve the issue on which they followed.
3. Further discussion of the obtained results and conclusions should be discussed. The presentation seems to be cut off.
4. To improve the reader's understanding of the significance of the proposed approach, two paragraphs/sections should be added to the text. Authors should devote one paragraph to a discussion of the limitations of the proposed approach and a second paragraph to its possible development.
Author Response
In the introduction, the current state is practically not discussed, but an analysis of publications devoted to the design of HBO is necessary.
Worthy reviewer, we did not discuss in detail about current state because current state mostly focuses on the production of mutant recombinant hemoglobin molecules and development of whole RBC from stem cells which are out of the scope of our research. The main aim was to produce RBC substitutes using less and available resources. We have used cheapest ways to produce RBC substitutes which may be the best option in countries where blood related emergencies are high due to low donation rates and resources are low.
The authors need to substantiate the choice of the way to solve the issue on which they followed.
Worthy reviewer, we have updated the content. (Line: 155-161)
Further discussion of the obtained results and conclusions should be discussed. The presentation seems to be cut off.
Worthy reviewer, we have updated manuscript by adding a separate conclusion section.
To improve the reader's understanding of the significance of the proposed approach, two paragraphs/sections should be added to the text. Authors should devote one paragraph to a discussion of the limitations of the proposed approach and a second paragraph to its possible development.
Worthy reviewer, we have added limitations and its further possible development in the conclusion section.
Round 2
Reviewer 2 Report
The manuscript may be accepted as is.